# The role of uncertainty intolerance in adjusting to long-term physical health conditions: A systematic review

**Benjamin Gibson**[1], **Benjamin A. Rosser**[2], **Jekaterina Schneider**[3], **Mark J. Forshaw**[4]*

**1** School of Applied Social Sciences, Faculty of Health and Life Sciences, De Montfort University, Leicester, United Kingdom, **2** School of Psychology, Faculty of Health, Liverpool John Moores University, Liverpool, United Kingdom, **3** Centre for Appearance Research, School of Social Sciences, College of Health, Science and, University of the West of England, Bristol, United Kingdom, **4** Department of Psychology, Edge Hill University, Ormskirk, United Kingdom

* Mark.Forshaw@edgehill.ac.uk

## Abstract

Long-term physical health conditions (LTPHCs) are associated with poorer psychological well-being, quality of life, and longevity. Additionally, individuals with LTPHCs report uncertainty in terms of condition aetiology, course, treatment, and ability to engage in life. An individual's dispositional ability to tolerate uncertainty—or difficulty to endure the unknown—is termed intolerance of uncertainty (IU), and may play a pivotal role in their adjustment to a LTPHC. Consequently, the current review sought to investigate the relationship between IU and health-related outcomes, including physical symptoms, psychological ramifications, self-management, and treatment adherence in individuals with LTPHCs. A systematic search was conducted for papers published from inception until 27 May 2022 using the databases PsycINFO, PubMed (MEDLINE), CINAHL Plus, PsycARTICLES, and Web of Science. Thirty-one studies ($N = 6{,}201$) met the inclusion criteria. Results indicated that higher levels of IU were associated with worse psychological well-being outcomes and poorer quality of life, though impacts on self-management were less clear. With the exception of one study (which looked at IU in children), no differences in IU were observed between patients and healthy controls. Although findings highlight the importance of investigating IU related to LTPHCs, the heterogeneity and limitations of the existing literature preclude definite conclusions. Future longitudinal and experimental research is required to investigate how IU interacts with additional psychological constructs and disease variables to predict individuals' adjustment to living with a LTPHC.

## Introduction

Increasing life expectancy around the globe has been accompanied by an increased risk of long-term illness [1] and multimorbidity [2]. Long-term physical health conditions (LTPHCs), such as chronic pain, diabetes, and cardiovascular disease, therefore warrant worldwide attention and response [3]. United Kingdom-based estimates from Scotland suggest that around 42% of people may be living with at least one LTPHC [4], while in the United States,

**Data Availability Statement:** This is a systematic review and therefore no new data was created. However, all relevant information (such as the

search strategy) are within the manuscript and its Supporting information files.

**Funding:** The author(s) received no specific funding for this work.

**Competing interests:** The authors have declared that no competing interests exist.

prevalence may extend to over half the population [5]. The resultant economic costs are considerable [6] and they increase with multimorbidity [7]. For the individual, LTPHCs can also threaten both quality [8, 9] and longevity of life [3].

LTPHCs typically require both lifestyle adaptation and self-management [10]. While these factors can provide aspects of personal control and influence, such health conditions often also pose challenges that are ill-defined, uncontrollable, and ultimately *uncertain*: chronic pain may defy clear medical explanation [11]; multiple sclerosis may follow an uncertain trajectory [12]; and epilepsy may unpredictably cause seizures [13]. Whereas uncertainty associated with acute illness may be resolvable, long-term conditions often require adjustment and acceptance of ongoing, unavoidable unknowns [14]. Individuals with LTPHCs qualitatively report uncertainty in terms of condition aetiology, course, treatment, and ability to engage in life [15]. Consequently, living with a LTPHC requires living with uncertainty.

Understanding the relationship between LTPHCs and one's ability to deal with uncertainty is important, given that individual differences in the experience of uncertainty have been shown to inform different cognitive, emotional, and behavioural responses relevant to healthcare and condition management [16]. An individual's ability to tolerate uncertainty is therefore likely to play a pivotal role in one's adjustment and self-management in relation to a LTPHC. Difficulty enduring the unknown is termed intolerance of uncertainty (IU) and represents a dispositional experience of uncertainty as aversive and unbearable [17]. Although those who are less tolerant of uncertainty are more likely to take efforts to control the situation or eliminate the uncertainty [18], such attempts may inadvertently create further issues. For example, IU has been found to be associated with frequent and rigid avoidance behaviours [19]. These responses, which are aimed at controlling and/or avoiding unwanted internal experiences, appear to be a consistent feature of multiple psychological difficulties [20]. Whilst the application of rigid avoidance behaviours may be reinforced by short-term relief, they come at a long-term cost in that they may maintain and exacerbate difficulty by restricting an individual's behavioural repertoire at the expense of engagement in personally valued areas of life [21]. Consequently, these strategies may paradoxically increase the unwanted experience one is seeking to avoid (e.g., Wenzlaff and Wegner [22]). Indeed, multiple reviews have collated a substantial body of evidence linking IU with a range of psychological difficulties including anxiety, depression, obsessive-compulsive challenges, and eating disorders [23–27], many of which co-occur alongside LTPHCs [31].

The direct relevance of IU to LTPHCs is less clearly established compared to the mental health literature, but existing evidence suggests the relationship warrants more attention. For example, higher IU has shown to predict lower quality of life in individuals with epilepsy [28], increased anxiety, depression, and 'handicap' in individuals with Ménière's disease [29], and greater stress and non-somatic symptoms of depression and lower emotional well-being in individuals with lung cancer [30]. However, findings are inconsistent. While Mitmansgruber et al. [31] found a correlational association between IU and some quality of life domains in individuals with cystic fibrosis, IU failed to demonstrate predictive capacity in regression analysis that also included resilience variables. Similarly, Wilson et al. [32] failed to demonstrate that IU may predict adherence or retention in care among people with HIV.

Current research into the relationship between IU and LTPHCs presents other unexpected findings as well. For example, Taha et al. [33] compared levels of IU between patients and healthy controls and found that women post-cancer treatment actually reported greater *tolerance* of uncertainty than did women who had never had a cancer experience. The authors argued that although women post-treatment faced the threat of cancer recurrence, the findings provided evidence that the 'trait characteristic' of IU may be uniquely changeable following a significant life event. However, cancer is somewhat unique as a LTPHC as it can go into

remission and may offer patients an emotional and physical respite (even if recurrence remains a possibility). Follow-up to this investigation is important, as this and similar findings could have implications for how we define and treat IU, even beyond the scope of LTPHCs. To address both this issue and the inconsistency of the literature more generally, there is therefore a need for collation of existing evidence to provide an overarching and comprehensive account that can aid interpretation.

While multiple reviews exist collating the literature on IU and psychological difficulties, research on uncertainty in healthcare has been criticised as fragmented and in need of unification [16, 34]. To the authors' knowledge, only one systematic review relevant to IU and physical health exists [35]; although this review focused on healthcare in general, rather than on LTPHCs specifically. While methodological quality of evidence was low, the review found that patients with lower uncertainty tolerance were at greater risk of distress and more likely to engage in avoidant coping strategies. These findings suggest that IU may exacerbate health-related concerns and encourage responses that compound, rather than resolve, difficulties. However, Strout and colleagues'[35] review search was conducted in 2015 and requires updating. Indeed, more recent empirical evidence exists. For example, Neville and colleagues' [36] longitudinal investigation involving 152 young people with chronic pain found that higher IU predicted subsequent increases in pain interference through increased fear of pain and catastrophic appraisal of pain. Consequently, evidence in this area may be growing in quantity and quality, and research post-2015 pertaining to IU and LTPHCs is yet to be synthesised and evaluated. As such, a contemporaneous and comprehensive review of the existing literature exploring the relevance of IU to LTPHCs is warranted. The importance of fully understanding IU's role in the experiences and outcomes associated with LTPHCs is further underscored by its potential therapeutic utility (e.g., Molton et al. [37]) and possible amenability to change [33].

## The current study

The discussed literature suggests that LTPHCs are often accompanied by uncertainty. Difficulty tolerating this experience may increase the challenges posed by LTPHCs and potentially threaten adjustment. While multiple systematic reviews exist demonstrating the relevance of IU to psychological difficulties [23–26], similar amalgamation of the literature relating to outcomes in LTPHCs is limited and in need of update. Consequently, the current review sought to systematically investigate the relationship between IU and health-related outcomes, including physical symptoms, psychological ramifications (e.g., anxiety, depression, quality of life), self-management, and treatment adherence in individuals with LTPHCs. Additionally, it aimed to investigate potential differences in IU levels between patients with LTPHCs and healthy controls, in order to examine whether increased experience of uncertainty associated with LTPHCs sensitises a person to become less tolerant of uncertainty. Based on the literature outlined above, we hypothesised that: (1) higher IU would be associated with poorer physical and mental health outcomes in individuals with LTPHCs; (2) higher IU would be associated with poorer self-management and lower treatment adherence in individuals with LTPHCs; and (3) levels of IU would be comparable between samples of individuals with and without LTPHCs.

## Methods

This review was conducted according to the Preferred Reporting Items for Systematic Reviews and Meta-Analyses statement (PRISMA [38]) and pre-registered on PROSPERO prior to commencement (ref no. [CONCEALED]).

## Data sources and search strategies

A systematic search was conducted for papers published from inception until 27 May 2022 using the databases PsycINFO, PubMed (MEDLINE), CINAHL Plus, PsycARTICLES, and Web of Science. Searches were not restricted based on language or date of publication. Boolean combinations of search terms related to IU and LTPHCs were used (see S1 Table). Reference sections of included articles were scanned to identify additional studies that met inclusion criteria. For the purposes of this review, we limited our definition of long-term conditions as *physical*, rather than psychological, while accepting that there are psychological comorbidities present in many individuals with LTPHCs and vice versa. These are conditions for which there is no effective cure, but for which amelioration and management are the core care approaches. The *NHS DoH Long Term Conditions Compendium of Information* [39] gives the following as key physical long-term conditions in terms of prevalence in the United Kingdom population: hypertension, diabetes, asthma, coronary heart disease, chronic kidney disease, hypothyroidism, stroke, chronic obstructive pulmonary disease, cancer, atrial fibrillation, heart failure, and epilepsy. These conditions differ from disabilities in many individuals for medical and socio-political reasons, although long-term conditions themselves can give rise to disabilities. As such, we excluded both disabilities and psychological conditions from this review.

## Study eligibility criteria

Papers were eligible for inclusion if they: (a) described samples with participants who had at least one LTPHC and (b) quantitatively assessed the direct or indirect effects of IU on one or more LTPHC and related outcomes. Papers were excluded if they: (a) described qualitative studies or reviews; (b) included participants with disabilities or psychological conditions rather than LTPHCs (e.g., long-term hearing loss or schizophrenia); (c) did not distinguish between participants with and without LTPHCs in their analyses; or (d) did not directly measure IU or did not use a validated IU measure (e.g., the Intolerance of Uncertainty Scale; IUS [40, 41]). Likewise, studies that measured a construct that relates to, but differs from, IU (e.g., intolerance of *illness*) were also excluded.

Notably, we adopted a more flexible approach regarding the presence of a LTPHC among participants diagnosed with cancer specifically. After an initial literature search, it became clear that most studies included patients at varying stages of diagnosis, treatment, and disease progression, and many did not distinguish between these categories in their analyses. Additionally, although cancer is by definition considered a LTPHC (see above), patients can be considered in remission and without an active cancer diagnosis after successful treatment, although multiple forms of cancer have a high chance of recurrence [42]. As such, we included studies that described participant samples with an active cancer diagnosis only or samples with mixed disease stage, but that were within five years post-cancer diagnosis. This time frame was chosen in line with findings from the reviewed literature (e.g., Jones et al. [43]), as well as studies showing that psychological distress may be greatest during this period [44]. However, we excluded studies that explicitly stated that none of the participants had clinical evidence of disease at the time of recruitment.

Finally, we examined any analyses that explored the role of uncertainty in LTPHCs, including correlations/associations between IU and health-related outcomes, the mediating and/or moderating effect of IU on health-related outcomes, and differences in IU between patients with LTPHCs compared to healthy participants.

## Study selection and data extraction

After running the search, titles and abstracts were screened against the above eligibility criteria. This procedure was followed by full-text screening to remove any further irrelevant papers, as

well as duplicates. Two authors (BG and JS) independently screened all papers and extracted data from the identified studies. The following data were extracted: (a) author(s) and year of publication; (b) study design; (c) IU measure; (d) sample size (% women); (e) participant age; (f) LTPHC; (g) outcomes related to IU (including effect sizes where available); and (h) study quality (global rating). For studies that described statistically significant outcomes, a *p* value < .05 was considered significant (unless corrected or otherwise statistically adjusted).

### Quality assessment

Quality assessment of included studies was carried out using the Quality Assessment Tool for Quantitative Studies, developed by the Effective Public Health Practice Project (EPHPP [45]). The EPHPP provides an overall methodological quality rating of 'strong' (no weak ratings), 'moderate' (one weak rating), or 'weak' (more than one weak rating). The ratings are based on selection bias, study design, confounders, blinding, data collection method, and withdrawals and dropouts.

The EPHPP was chosen because it is suitable for evaluating the methodological quality of various study designs [46]. Additionally, it has been found to have excellent inter-rater reliability for overall scores when compared to the Cochrane Collaboration Risk of Bias Tool [47, 48] and established construct and content validity [46]. Two authors (BG and JS) independently assessed all studies. Cohen's kappa [49] was calculated to determine inter-rater reliability, showing moderate agreement (87.1%) between scores ($\kappa$ = .798, *p* < .001). Discrepancies were due to differences in interpretation of criteria (particularly related to selection bias) and were discussed among the authors until a 100% agreement in coding was reached.

## Results

As at 27 May 2022, the search protocol yielded 833 papers (see Fig 1). After removing duplicates and non-relevant results, 341 papers were screened and 99 reports were sought for retrieval, of which 20 reports could not be retrieved (e.g., searches provided incomplete/ inaccurate references or access to the full text was restricted). In total, 79 articles were assessed for eligibility. Seven studies were excluded because they did not measure IU or did not use a validated IU measure, thirty-five studies were excluded because they did not assess the relationship between IU and LTPHC outcomes, four studies were excluded because they did not distinguish between individuals with and without a LTPHC in their analyses, and two studies were excluded because participants did not have an active LTPHC at time of recruitment.

### Study characteristics

A final sample of 31 studies (6,201 participants) was included in this review (see Table 1). The majority of the included studies were cross-sectional in design (*n* = 20), with five longitudinal studies, five case-control studies (i.e., studies that compared patients with healthy controls at one point in time), and one randomised controlled trial. LTPHCs included various forms of cancer (*n* = 12), multiple sclerosis (*n* = 2), Ménière's disease (*n* = 2), HIV (*n* = 2), congenital heart disease (*n* = 2), chronic pain (*n* = 2), and one study each for Parkinson's disease, Crohn's disease, epilepsy, hypertension, cystic fibrosis, type 2 diabetes, irritable bowel syndrome, inflammatory bowel disease, and Lynch syndrome. Participants were predominantly adults, with one study conducted among younger adults (i.e., university students) and three with children and/or adolescents. The majority of studies used various versions of the IUS to measure intolerance of uncertainty (*n* = 28), while three studies used tolerance of ambiguity or tolerance of uncertainty subscales.

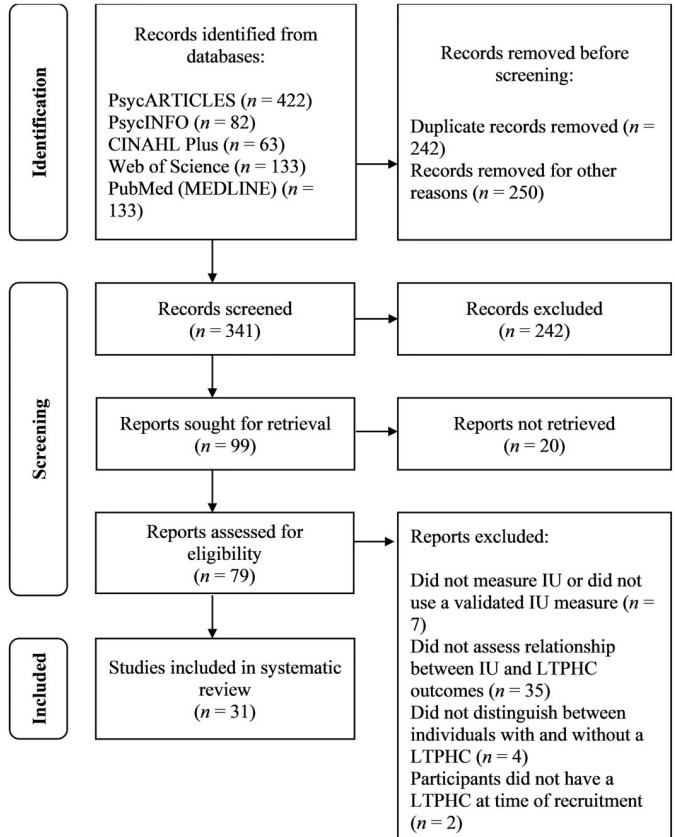

**Fig 1. PRISMA flowchart of study selection.**

## Study quality

In terms of study quality, nine studies were rated as 'weak', sixteen studies were rated as 'moderate', and six studies were rated as 'strong'. Most studies lacked quality in study design and/or selection bias. Study design concerns the likelihood of bias due to the allocation process in experimental studies or the extent that assessments of exposure and outcome are likely to be independent in observational studies. Selection bias, on the other hand, considers how representative the sample is of the target population and the percentage of approached participants that agree to take part in the study. As such, all cross-sectional studies were rated as 'weak' for study design, longitudinal and case-control studies were rated as 'moderate', and one study adopting a randomised controlled trial was rated as 'strong'. In terms of selection bias, studies received a 'strong' score if the selected individuals were very likely to be representative of the target population (e.g., randomly selected from a comprehensive list of individuals in the target population) and there was greater than 80% participation; a 'moderate' score if the selected individuals were somewhat likely to be representative of the target population (e.g., referred from a source or clinic) and there was 60–79% participation; and a 'weak' score if the selected individuals were not likely to be representative of the target population (e.g., self-referred) and there was less than 60% participation, or if the selection and level of participation were not described.

**Table 1. Characteristics of studies included in systematic review.**

| Author(s) (Year) | IU Measure | Sample Size $N$ (% Women) | Age in Years $M$ ($SD$) | LTPHC | Results | Study Quality |
|---|---|---|---|---|---|---|
| *Cross-Sectional Studies* | | | | | | |
| Apolinário-Hagen et al. (2018) | 4-item IUS | 98 (67.3) | 47.0 (10.2) | Multiple sclerosis | IU positively associated with acceptance of mHealth apps and predicted intention to use mHealth apps for the management of multiple sclerosis, mediated by self-efficacy ($B$ = -.095, 95% CIs: -.227, -.01) | Weak |
| Barahmand & Haji (2014) | 27-item IUS | 60 (53.3) | 33.1 (12.4) | Epilepsy | IU negatively associated with QoL ($r$ = -.438, $p$ < .001) and positively associated with worry ($r$ = .462, $p$ < .001) and irritability ($r$ = .622, $p$ < .001); irritability mediated relationship between IU and QoL ($B$ = .067, 95% CIs: -.07, .21, β = .17, $t$ = .972, $p$ = .338) | Moderate |
| Brown & Fernie (2015) | 27-item IUS | 106 (28.3) | 65.6 (9.3) | Parkinson's disease | IU positively associated with anxiety ($r$ = .55, $p$ < .001) and distress ($r$ = .38, $p$ < .001); severity of symptoms not associated with IU ($p$ > .05) | Weak |
| Cohen et al. (2022) | 12-item IUS | 93 (46.2) | 63.2 (13.8) | Cancer (various) | IU positively associated with psychological distress ($r$ = .34; β = .34, $p$ < .01), which was partially mediated by perceived COVID-19 threat and impact on health (β = .18, 95% CIs: .07, .32, $F$(3,89) = 10.23, $p$ < .001, $R^2$ = .26) | Moderate |
| Costa-Requena et al. (2011) | 27-item IUS | 26 (100) | 53.1 (1.1) | Breast cancer | IU predicted depression ($F$ = 6.86, $p$ = .016) and worry ($F$ = 7.15, $p$ = .015), but not anxiety ($F$ = 3.13, $p$ = .092) | Moderate |
| Curran et al. (2020) | 12-item IUS | 211 (83.9) | 60.3 (10.9) | Cancer (various) | IU positively associated with fear of cancer recurrence ($r$ = .51, $p$ < .001), but did not predict fear of recurrence in multivariate models | Moderate |
| Drews & Hazlett-Stevens (2008) | 27-item IUS | 391 (66.2) | 19.5 (3.7) | Irritable bowel syndrome | No significant differences in IU between participants with irritable bowel syndrome and those without following Bonferroni correction of α < .01 ($t$(355) = -1.99, $p$ < .047) | Weak |
| Eisenberg et al. (2015) | 27-item IUS | 67 (0) | 64.3 (8.0) | Prostate cancer | IU positively associated with cancer-related distress (β = .34, $t$(61) = 3.06, $p$ = .003), avoidance (β = .36, $t$(61) = 2.85, $p$ = .006), and hyperarousal (β = .30, $t$(61) = 2.53, $p$ = .014) after adjusting for age, education, fear of recurrence, cancer-related physical symptoms, and cognitive complaints; IU moderated relationship between cognitive complaints and intrusive thoughts | Moderate |
| Hill et al. (2021) | 12-item IUS | 100 (100) | 55.0 (12.0) | Ovarian cancer | IU positively associated with anxiety ($r$ = .497, $p$ < .01), stress ($r$ = .567, $p$ < .01), and depressive symptoms ($r$ = .437, $p$ < .01) | Weak |
| Hill & Hamm (2019) | 12-item IUS | 131 (100) | 52.5 (10.0) | Ovarian cancer | IU positively associated with depressive ($r$ = .403, $p$ < .01) and anxiety symptoms ($r$ = .445, $p$ < .01), and negatively associated with social support ($r$ = -.330, $p$ < .01) | Weak |
| Jones et al. (2014) | 12-item IUS | 137 (100) | 49.1 (10.6) | Breast cancer | IU positively associated with health anxiety ($r$ = .50, $p$ < .001), but did not predict health anxiety in multiple regression analysis | Weak |
| Kurita et al. (2013) | 27-item IUS | 49 (71.4) | 64.2 (11.0) | Lung cancer | IU positively associated with stress, poorer emotional well-being, and depressive symptoms; avoidance fully mediated relationship of IU with depressive symptoms (path c–path c' = .08, 95% CIs: .004, .24) and emotional well-being (path c–path c' = .06, 95% CIs: .17, .02), but not with stress (path c–path c' = .06, 95% CIs: -.004, .21) | Weak |

(*Continued*)

**Table 1.** (Continued)

| Author(s) (Year) | IU Measure | Sample Size N (% Women) | Age in Years M (SD) | LTPHC | Results | Study Quality |
|---|---|---|---|---|---|---|
| Lebel et al. (2018) | 27-item IUS | 106 (100) | 56.8 (10.6) | Breast or gynaecological cancer | IU positively associated with fear of cancer recurrence ($r = .31$, $p < .001$), but did not predict fear of recurrence in multivariate models ($\beta = .17$, $p = .07$); IU predicted maladaptive coping strategies ($\beta = .24$, $p < .05$) | Moderate |
| Llewelyn-Williams et al. (2022) | 12-item IUS-R (revised for school-aged children) | Young people: 36 (45.7) Parents: 35 (89.0) | Young people: 10.5 (IQR = 4.0) Parents: 44 (IQR = 10.5) | Congenital heart disease | Associations observed between young people's IU and parent state ($r = .37$, 95% CIs: .052, .626) and trait anxiety ($r = .46$, 95% CIs: .157, .686) but not between young people's IU and their own health anxiety | Moderate |
| López-Martínez et al. (2022) | 12-item IUS | 188 (83.5) | 59.9 (10.1) | Chronic pain | IU moderated association between anxiety and catastrophizing ($B = .039$, $SE = .012$, 95% CIs: .015, .063), and between catastrophizing and pain intensity ($B = -.034$, $SE = .010$, 95% CIs: -.054, -.014); anxiety and IU did not interact in predicting catastrophizing ($B = .004$, $SE = .002$, 95% CIs: -.008, .000), although an interaction effect was found between IU and catastrophizing in predicting pain intensity ($B = .010$, $SE = .005$, 95% CIs: .001, .019) | Moderate |
| López et al. (2008) | SRSS-12 (tolerance to ambiguity subscale) | 64 (39.1) | 36.9 (-) | HIV | Tolerance of ambiguity and stress predicted adherence to treatment ($\beta = .399$) | Weak |
| Miles et al. (2020) | 4-item IUS | 129 (40.3) | Median 66.4yrs | Known or suspected colorectal or lung cancer | IU predicted psychological distress regardless of known or suspected diagnosis ($OR = 2.231$, 95% CIs: 1.429, 3.485, $p < .001$) | Weak |
| Mitmansgruber et al. (2016) | 18-item IUS | 57 (45.6) (multiple healthy reference groups) | 28.5 (range 18-58yrs) | Cystic fibrosis | No significant differences in IU compared to healthy controls ($n = 540$ students, $p > .05$); stress due to IU negatively associated with QoL ($p < .05$) | Moderate |
| Sagarduy et al. (2018) | SRSS-12 (tolerance to ambiguity subscale) | 182 (76.4) | 59.6 (9.9) | Hypertension | Greater tolerance to ambiguity had a positive effect on physical activity behaviour ($\beta = .24$, 95% CIs: .00, .45, $p = .049$) | Moderate |
| Torbit et al. (2016) | 12-item IUS | 128 (100) | 52.5 (14.5) | Lynch syndrome | IU positively associated with anxiety ($r = .388$, $p < .01$), depression ($r = .315$, $p < .01$), and worry interference ($r = .333$, $p < .01$) | Moderate |
| *Longitudinal Studies* | | | | | | |
| Kirby & Yardley (2009b) | 27-item IUS | 358 (68.7) | Range 28-90yrs | Ménière's disease | IU positively associated with anxiety at baseline ($F = 85.89$, $p < .001$, $d = 1.01$) and at 3 months ($F = 69.89$, $p < .001$, $d = .88$); IU predicted anxiety at 3 months ($B = 0.05$, $SE = .01$, Wald statistic = 20.54, $p < .001$) | Strong |
| Neville et al. (2021) | 12-item IUS-R | 152 (72.4) | 14.23 (range 10-18yrs) | Chronic pain | IU had an indirect effect on 3-month pain interference via youth pain catastrophizing and fear of pain ($b = .132$, 95% CIs: .078, .198, $p = .009$) | Strong |
| Stone et al. (2022) | 12-item IUS | 154 (69.5) | 43.4 (12.5) | Inflammatory bowel disease | IU was not associated with various indices of active disease after adjusting for other factors, with the exception of lower self-reported flares ($OR = .93$, 95% CIs: .87, .99) | Strong |
| Tan et al. (2016) | 8-item IUS | 119 (0) | - | Prostate cancer | IU positively associated with generalised ($OR = 1.22$, 95% CIs: 1.09, 1.38) and prostate cancer specific anxiety ($OR = 1.29$, 95% CIs: 1.13, 1.49) | Moderate |

(*Continued*)

**Table 1.** (Continued)

| Author(s) (Year) | IU Measure | Sample Size *N* (% Women) | Age in Years *M* (*SD*) | LTPHC | Results | Study Quality |
|---|---|---|---|---|---|---|
| Wilson et al. (2018) | HCEI (tolerance of uncertainty subscale) | 973 (100) | 49.3 (8.5) | HIV | Tolerance for uncertainty did not predict adherence (β = .03, *SE* = .02, 95% CIs: -.002, .070) or retention in care (β = .009, *SE* = .009, 95% CIs: -.006, .030) | Strong |

*Case-Control Studies*

| Author(s) (Year) | IU Measure | Sample Size *N* (% Women) | Age in Years *M* (*SD*) | LTPHC | Results | Study Quality |
|---|---|---|---|---|---|---|
| Kirby & Yardley (2009a) | 27-item IUS | Patients: 800 (63.1) Healthy controls: 484 (55.4) | Patients: 60.5 (12.5) Healthy controls: 55.6 (14.4) | Ménière's disease | IU positively associated with anxiety (*r* = .66) and depression (*r* = .54); PTSD symptoms mediated relationship of IU with depression (*Aroian* = 15.61, *p* < .001) and handicap (*Aroian* = 14.12, *p* < .001); no significant differences in IU compared to healthy controls (*p* > .05) | Moderate |
| Oliver et al. (2018) | 12-item IUS-R | 84 (42.9) (42 patients, 42 healthy controls) | 11.7 (2.5) | Congenital heart disease | Children and adolescents with congenital heart disease demonstrated significantly higher IU (*F* (1, 81) = 6.36, *p* = .014, η$p^2$ = .07); IU positively associated with health anxiety among healthy controls only (*r* = .48, *p* < .001) | Strong |
| Rasmussen et al. (2013) | 12-item IUS | 312 (53.8) | 62.4 (14.1) | Type 2 diabetes | No significant differences in IU between patients with high HbA1c, patients with acceptable HbA1c, or healthy controls (*p* = .11) | Moderate |
| Rubio et al. (2016) | 27-item IUS | Patients: 9 (66.7) Healthy controls: 9 (44.5) | Patients: 41.0 (3.0) Healthy controls: 34.0 (4.0) | Crohn's disease | No significant differences in IU compared to healthy controls (*p* = .48); patients showed significantly increased brain responses to uncertainty regarding upcoming uncomfortable rectal distension; brain responses to uncertainty and anticipatory fear levels proportionate to levels of trait anxiety, IU, and hypervigilance regarding visceral sensations | Strong |
| Salamanca- Balen et al. (2021) | 12-item IUS | Patients: 155 (71.0) Healthy controls: 150 (71.0) | Patients 54.3 (13.4) Healthy controls 54.1 (12.7) | Cancer (various) | IU had an indirect effect on the stress-emotional well-being relationship in both cancer (*B* = -.011, 95% CIs: -.020, -.003) and non-cancer groups (*B* = -.012, 95% CIs: -.024, -.001), but an indirect effect on the stress-physical well-being relationship in the non-cancer group only (*B* = -.008, 95% CIs: -.017, -.001) | Moderate |

*Randomised Controlled Trials*

| Author(s) (Year) | IU Measure | Sample Size *N* (% Women) | Age in Years *M* (*SD*) | LTPHC | Results | Study Quality |
|---|---|---|---|---|---|---|
| Molton et al. (2019) | 27-item IUS | 48 (72.9) | 37.9 (10.9) | Multiple sclerosis | Improvements in ability to tolerate uncertainty were associated with decreases in global anxiety (*r* = .54, *p* < .05) and increases in multiple sclerosis acceptance (*r* = -.63, *p* < .01) | Moderate |

HCEI = Health Care Empowerment Inventory; HIV = Human immunodeficiency virus; IU = Intolerance of uncertainty; IUS = Intolerance of Uncertainty Scale; LTPHC = Long-term physical health condition; PTSD = Post-traumatic stress disorder; QoL = Quality of life; SRSS-12 = Stress-Related Situations Scale.

## Hypothesis 1: Association of intolerance of uncertainty with mental and physical health outcomes

With the exception of one paper [50], all of the included studies found an association between IU and psychological well-being. Specifically, IU was positively associated with anxiety, stress, depressive symptoms [29, 30, 37, 43, 51–57], fear of pain or illness recurrence [36, 58, 59], worry or worry interference [28, 57], irritability [28], pain interference [36], 'handicap'[29], and psychological distress [51, 60–62]. Moreover, IU was negatively associated with quality of life [28, 31]. Contrary to predictions, one study found no association between IU and

children's health anxiety among children with congenital heart disease, despite evidence that IU was positively associated with parents' state and trait anxiety [50].

Several studies examined the role of IU alongside other constructs, to examine how variables interacted with IU to influence target outcomes (e.g., anxiety, fear of pain), and whether other factors helped explain the relationship of IU with psychological outcomes in individuals with LTPHCs. Multiple regression analyses, which included IU alongside other constructs such as metacognitions and illness-related anxiety, produced mixed results. Eight studies reported a significant direct relationship between IU and psychological outcomes of interest [29, 36, 51, 53, 55–57, 62], while five studies reported limited or no predictive power of IU in multivariate analyses [31, 43, 58, 59, 63]. Among these five studies, individual variables (e.g., younger age, resilience), disease variables (e.g., higher stage of cancer, disease duration), and variables directly relevant to the LTPHCs and outcomes of interest (e.g., body vigilance, threat appraisal) were found to have a stronger relationship with psychological well-being than IU. Notably, of the above 13 studies, only four were longitudinal [36, 55, 56, 63]. These studies represented evidence of either 'moderate' or 'strong' quality and all but one [63] showed a positive direct effect of IU on psychological difficulty outcomes.

Moreover, multiple studies found that the relationship between IU and target outcomes was partially or fully mediated by other psychological constructs. Barahmand and Haji [28] found that the relationship between IU and quality of life was mediated by worry and irritability in persons with epilepsy. Kurita et al. [30] found that avoidance fully mediated the relationship of IU with depressive symptoms and emotional well-being in patients with lung cancer. Neville et al. [36], meanwhile, found that IU had an indirect effect on 3-month pain interference via pain catastrophising and fear of pain in youth with chronic pain. Comparably, Kirby and Yardley [29] found that although IU was directly associated with anxiety, its association with depression and 'handicap' was mediated by post-traumatic stress disorder (PTSD) symptoms.

One study examined IU's role as a mediator between related variables and psychological well-being. Salamanca-Balen et al. [64] found that IU had an indirect effect on the stress and emotional well-being relationship in both patients with cancer and healthy controls during the uncertain period associated with the COVID-19 pandemic. Interestingly, however, the authors only identified an indirect effect of IU on the stress and physical well-being relationship among healthy controls. A further five studies explored IU as a potential moderator. Specifically, Eisenberg et al. [61] found that IU moderated the relationship between cognitive complaints and intrusive thoughts in prostate cancer survivors, and Torbit et al. [57] found that the interaction of IU and trust in one's physician were significantly associated with greater cancer worry interference in patients with Lynch syndrome. Similarly, López-Martínez et al. [65] found that IU moderated the association between anxiety and catastrophising, and between catastrophising and pain intensity in patients with chronic pain. In contrast, Hill and Hamm [54] found no interaction effects between IU and social support or loneliness in predicting depressive and anxiety symptoms in patients with ovarian cancer. Thus, although findings suggest that IU is negatively associated with psychological well-being in patients with LTPHCs, results are currently limited regarding the interaction between IU and additional constructs that may combine to explain a greater variance in target outcomes.

Notably, only one study examined the relationship between IU and physical health outcomes. In contrast to our expectations, Mitmansgruber and colleagues [31] found no association between IU and lung function or body mass index among people with cystic fibrosis. Consequently, our first hypothesis was partially supported.

## Hypothesis 2: The role of intolerance of uncertainty in self-management and treatment adherence

Regarding our second hypothesis, few studies examined the relationship between IU and self-management or treatment adherence in individuals with LTPHCs. Furthermore, the definition of self-management and how it was measured varied across studies. Greater tolerance to ambiguity predicted adherence to treatment among individuals positive for HIV [66] and was found to have a positive effect on physical activity behaviour for the management of hypertension [67]. Conversely, IU predicted an increase in the use of maladaptive coping strategies (such as cognitive avoidance) in patients with cancer [59]. Notably, Apolinário-Hagen et al. [68] found that IU was positively associated with behavioural intention to use mHealth apps for the management of multiple sclerosis in simple regression analysis, mediated by computer self-efficacy and multiple sclerosis self-efficacy. However, the effect of IU was no longer significant in a multiple regression model, with only performance expectancy and social influence remaining significant predictors of the intention to use mHealth apps. Contrary to our predictions, Wilson et al. [32] found a non-significant effect of IU on viral control/load suppression (as measured by adherence and retention of medication) in HIV care. Most of the above studies represented a 'weak' to 'moderate' level of evidence, while Wilson et al. [32] represented a 'strong' level of evidence. Our second hypothesis was thus partially supported.

## Hypothesis 3: Differences between patients with long-term physical health conditions and healthy controls

Several studies investigated differences in IU between patients with a LTPHC and participants without LTPHCs. In line with our third hypothesis, the majority of the identified studies found no significant differences in IU between patients and healthy controls among samples with Crohn's disease [69], cystic fibrosis [31], type 2 diabetes [70], Ménière's disease [29], and irritable bowel syndrome [71]. However, one study found that children and adolescents with congenital heart disease demonstrated significantly higher IU scores than healthy participants [72]. The majority of the above studies were rated 'moderate', with one study rated as 'weak' [71] and one study representing a 'strong' level of evidence [72]. As such, our third hypothesis was partly supported.

## Discussion

The aim of this review was to investigate the role of IU in patient outcomes, condition self-management, and treatment adherence in individuals with LTPHCs. LTPHCs are often characterised by great uncertainty; uncertainty around life expectancy, treatment, development or worsening of symptoms, quality of life, the needs and ability to make behaviour modifications, and prospects of disability (see, for example, Dauphin et al. [73]; Furlotte and Schwartz [74]; and Middleton et al. [75]). For those with dispositional difficulty tolerating uncertainty, navigating life with a LTPHC may be especially difficult and could lead to negative consequences for psychological and physical health [16]. In conducting this review, we expected that: (1) higher levels of IU would be associated with poorer physical and mental health outcomes in individuals with LTPHCs; (2) higher levels of IU would be associated with poorer self-management and treatment adherence; and (3) IU levels would be comparable between individuals with and without LTPHCs. All three of our hypotheses were partially supported by the existing evidence, yet there were several contradictions and nuances that warrant discussion.

## The relevance of intolerance of uncertainty to physical and mental health outcomes

The reviewed evidence was predominantly consistent with our first hypothesis, although some notable discrepancies and limitations were observed. Participants with a wide range of LTPHCs demonstrated consistent associations between greater IU and poorer mental health outcomes (e.g., anxiety, stress, depressive symptoms, worry or worry interference, irritability, and psychological distress) and lower overall quality of life. These findings are perhaps unsurprising given the volume of evidence demonstrating associations between IU and a range of psychological difficulties [23–26]. Additionally, the reviewed evidence also suggests that IU may demonstrate association with health-specific psychological processes, including pain catastrophising [36], fear of pain [36], fear of cancer recurrence [58, 59], and condition-related 'handicap' [29]. These findings suggest that IU's relevance to psychological outcomes therefore extends to populations with LTPHCs, in terms of both mental health and condition-specific outcomes.

Notably, extremely limited data were available on the direct relationships between IU and physical health outcomes [31] and no significant associations were found. Previous research has found associations between related psychological constructs (such as anxiety, depression, and illness-specific distress) with physical health outcomes such as HbA1c in people with type 2 diabetes [76] and risk of mortality in people with heart failure [77], so future research is required to more comprehensively assess the effect of IU on physical health outcomes.

While the majority of the reviewed studies reported significant associations between IU and mental health outcomes, even when considered alongside other predictive variables, five studies did not [31, 43, 58, 59, 63]. Instead, these studies indicated the relevance of other factors beyond IU (e.g., age, illness stage, threat appraisal, and resilience factors [personal competency and acceptance of life and self]) in predicting target outcomes. Notably, the combined studies provide nascent evidence that IU may moderate (and partially mediate [65]) other psychological processes within the context of LTPHCs, such as cognitive complaints [61] and trust in one's physician [57]. As the experience of LTPHCs cannot be reduced to an experience of uncertainty, it is important that the relative and interactive contribution of IU to outcomes is further clarified.

Furthermore, few studies explored the mechanisms by which IU may contribute to psychological outcomes. However, the reviewed studies provide some evidence that the relationship between IU and these outcomes may be mediated at least partly through processes such as worry and irritability [28], condition-related fear and catastrophising [36], and avoidance [30]. This is in line with theoretical assumptions around avoidance strategies associated with IU outlined in the introduction of this paper [19, 20]. An additional study reported that PTSD symptoms mediated the relationship between IU and both depression and 'handicap' in individuals with Ménière's disease [29]. The overall picture is consistent with the proposition that individuals who find uncertainty intolerable may be particularly threatened by the experience of having a LTPHC and fearful of its implications [17, 78], and thus engage in behaviours and cognitions aimed at controlling, reducing, and/or avoiding uncertainty [79]. However, bar one exception [36], all reviewed studies that conducted mediation analyses utilised solely cross-sectional data, which are insufficient to provide a full assessment of mediation and causality [79, 80]. Indeed, few of the reviewed studies involved designs including any assessment across time [36, 37, 55, 56].

## The relevance of intolerance of uncertainty to self-management and treatment adherence

Our second hypothesis was provisionally supported, although the available evidence investigating IU's association with LTPHC self-management and treatment adherence was limited,

which precludes us from making any definitive conclusions. The reviewed studies demonstrated associations between greater ambiguity tolerance and better self-management and treatment adherence [66, 67], whereas greater IU predicted use of maladaptive coping strategies [59]. The evidence suggests that difficulties coping with uncertainty may pose a barrier to self-management and treatment adherence, perhaps because such actions are in conflict with the avoidant strategies associated with higher IU. Apolinário-Hagen and colleagues' [68] finding of a positive association between IU and intention to engage with mHealth apps for the management of multiple sclerosis is also arguably consistent with this interpretation. While the authors suggest that this finding may unexpectedly represent IU as a facilitator of adaptive coping strategies, health monitoring is not exclusively adaptive. Monitoring may be motivated by attempts to avoid or resolve uncertainty, and these aims may not be achievable or otherwise serve the individual. For instance, IU may both drive increased health information seeking [81] and make individuals more vulnerable to anxiety when uncertainty is unresolvable [19]. Repeated health monitoring may even be considered ruminative. Indeed, IU itself has been linked with rumination [82], which is associated with negative outcomes (e.g., psychological distress) in individuals with LTPHCs [83, 84]. Consequently, in the context of LTPHCs, where it may be adaptive to develop acceptance and assimilation of the ongoing uncertainty associated with the health condition [14], IU may potentially disrupt adjustment and condition management in various ways.

However, the review identified one study that did not align with the proposition above. Wilson et al. [32] found no association between IU and subsequent viral control in women diagnosed with HIV. The authors suggest that the duration since diagnosis may explain this finding, as many participants had lived with HIV for numerous years. The influence of IU depends on the presence of uncertainty. Where sufficient condition management is possible and can provide stability, uncertainty may be most pronounced earlier in the condition course and may potentially diminish as the individual gains more experience of living with the LTPHC. As Wilson et al. [32] suggest, the effectiveness of modern HIV interventions (e.g., Pre-Exposure Prophylaxis [85] and Antiretroviral Therapies [86]), may help reduce the uncertainty experienced by individuals living with HIV. However, elsewhere, notable uncertainties have been reported to remain within this population [87]. Consequently, while conclusions cannot be drawn from this single study, the questions raised are notable and echo the suggestions of other authors regarding the potential relevance of duration from diagnosis and treatment stage [43]. There is a clear need for research considering the temporal impact of IU across the course of LTPHCs. Furthermore, more research is required to explore how IU relates to specific self-management and treatment adherence behaviours across various LTPHCs.

### Intolerance of uncertainty levels in individuals with and without long-term physical health conditions

Consistent with our third hypothesis, all but one of the reviewed studies found no significant differences in IU between patients and control participants. This finding was evident across a range of LTPHCs (e.g., Crohn's disease, cystic fibrosis, type 2 diabetes, Ménière's disease, and irritable bowel syndrome). Consequently, the presence of a LTPHC does not appear to universally elevate IU, which is consistent with the conceptualisation of IU as a stable trait [17]. However, one of the reviewed studies [72] did observe higher levels of IU in individuals with congenital heart disease compared to healthy controls, though it should be noted that participants in this study were children and adolescents. Speculatively, it is plausible that given the formative nature of childhood in the development of one's understanding of self, others, and

the world [88], such differences demonstrate a developmental period where IU is particularly malleable and susceptible to the impact of LTPHCs. Indeed, experience of unpredictability in childhood may predict unhelpful schematic representations held in adulthood [89]. While a recent meta-analysis suggests stable associations between IU and symptoms of psychological difficulties across the lifespan [24], this unique finding raises the question of whether LTPHCs could potentially contribute to determining the level of IU in childhood and beyond.

Indeed, Taha et al. [33] (though notably excluded from our review as an undefined number of participants likely had been cancer-free for 5+ years) had previously suggested that significant life events such as LTPHCs may alter IU levels. However, whereas Taha et al. [33] found that surviving breast cancer seemingly *reduced* IU compared to healthy controls (perhaps as part of Post-Traumatic Growth; PTG [90]), it is possible that Oliver et al.'s [72] study represents a similar finding, given that they may have demonstrated what happens when particularly vulnerable people are unable to cope or adapt. However, without further evidence and longitudinal assessment to draw upon for comparison, these interpretations remain speculative.

## Implications for intervention

This review provides preliminary (and primarily correlational) evidence of the involvement of IU in patient outcomes and response to LTPHCs. Consistent with critique of the mental health literature [26], more evidence is needed to substantiate whether IU exerts a causal influence. However, if substantiated, IU presents a psychological construct that is potentially amenable to change (e.g., Molton et al. [37]; also see Rosser [26] for examples), despite being conceptualised as a trait characteristic. Such traits may be useful indicators of treatment foci [91], precisely because they indicate areas of consistent difficulty and thus may inform selection of potentially impactful therapeutic targets. Consequently, for individuals experiencing high IU, increasing one's ability to tolerate uncertainty may present a useful therapeutic aim. Even if IU remains stable, intervention may still help individuals consider more adaptive coping strategies than those currently utilised to improve various outcomes [33, 59].

For example, Cognitive Behavioural Therapy (CBT) specifically targeting IU has demonstrated reductions in distress associated with IU as well as other symptoms of psychological difficulty [92–94], and so too has CBT even without a specific IU focus [95, 96]. Crucially, these interventions may remain relevant within the context of LTPHCs. For instance, Molton et al. [37] piloted an intervention focused on managing uncertainty in individuals with multiple sclerosis using techniques based on traditional cognitive-behavioural principles and Acceptance and Commitment Therapy (ACT [21]). The intervention lowered distress associated with IU, which consequently increased condition acceptance. Elsewhere, ACT has also been shown to decrease IU-related distress and psychological difficulties in individuals with type 2 diabetes [97], while early pilot studies suggest that making modifications to ACT in the future may likewise reduce IU-related distress in people with cancer [98]. Furthermore, mindfulness has also been shown to decrease prostate cancer anxiety and uncertainty intolerance, while increasing global mental health and PTG in prostate cancer survivors [99]. Overall, evidence is nascent but suggests that a range of existing therapeutic approaches may hold utility in supporting individuals with LTPHCs that report high IU.

## Strengths, limitations, and future directions

This review represents a contemporaneous and comprehensive collation of the literature exploring the relevance of IU to LTPHCs that has been needed since Strout and colleagues [35] first published a review of IU's impact on healthcare more generally. The strengths of the

review include: (1) extending consideration of IU to the context of physical healthcare; (2) systematic collation of the current evidence base; (3) inclusion of a broad range of LTPHCs and study designs; and (4) consideration of clinical application. However, several limitations should also be acknowledged.

First, although we aimed to address the quantitative evidence in this field, qualitative methods may provide additional, valuable understanding of the role of IU in LTPHCs. Qualitative methods not only provide intensive examination of a phenomenon but have become increasingly used in health psychology research over the last few years [100]. Including qualitative papers, therefore, may have allowed us to not only identify further relevant work pertaining to LTPHCs not identified in our search but to also provide deeper insight into people's experiences of IU in this context. For example, research by Brown and colleagues [15] showed that uncertainty could lead to frustration in patients with various LTPHCs who too often felt dependent on their doctor or healthcare professional to make things better. Even when symptoms were controlled, fear of the illness worsening and treatment failing remained, though there was also evidence that participants were able to develop effective coping strategies to combat this, such as by finding a routine, researching symptoms on their own, exercising, and planning ahead. As such, future reviews may wish to employ mixed-method approaches to provide more holistic insights regarding the influence of IU on the experience of living with LTPHCs. Furthermore, quantitative researchers could also include more detailed assessment of health conditions, prognosis and treatment, and other health information to likewise improve the depth and quality of the data.

Second, 12 of the 31 reviewed studies included participants with some form of cancer, which may have biassed the results. As outlined previously, certain types of cancer may present unique considerations in terms of the possibility of recovery/remission and the probability of recurrence [58, 59], which may cause a differential impact on the uncertainty experienced in contrast to other LTPHCs. Third, the heterogeneity of reviewed evidence prohibited meta-analysis. Future reviews may be better positioned to provide such statistical overview. Fourth, this review focused only on studies including validated self-report measures of IU. This approach aimed to enhance the reliability of evidence compiled and conclusions drawn; however, we acknowledge the calls for IU research to extend to more varied assessment methods to overcome the limitations of reliance on self-report [33, 78]. As use of such methods grows, future reviews may seek to incorporate behavioural and self-report assessment of IU to provide a more comprehensive overview. Finally, the review included intolerance of ambiguity alongside IU due to the theoretical overlap between these constructs and to cast a broad net given the paucity of existing research in this area. Only two reviewed studies focused on tolerance of ambiguity [66, 67] and their results were consistent with related IU evidence [59] and our hypotheses. However, we acknowledge that these constructs may be considered distinct: Grenier and colleagues [101] propose that IU has a future-focus whereas intolerance of ambiguity relates to ability to cope with unknowns in the present. Future research may consider distinguishing between these related constructs.

The current review also highlights limitations within existing literature that may direct future research. First, representation of the wide range of LTPHCs can be improved. Many health conditions are not currently represented in the literature. Existing research has predominantly involved participants with cancer. Even among traditionally well-researched conditions, such as chronic pain, where one may reasonably suspect an impact of IU (e.g., the vast majority of cases of lower back pain are of unknown cause [102]), there is a dearth of investigations into IU's relationship with patients' mental and physical health. Second, despite the prevalence and impact of multiple LTPHCs, particularly in older age [2, 8], multimorbidity was not considered in the studies identified by this review. Consequently, more research is

required on a wider range of conditions and with consideration of multimorbidity. Third, this review highlights that there is a serious need for more research into IU's relationship with self-management and treatment adherence. The reviewed studies included a very limited focus of self-management and adherence (i.e., adherence to HIV medication, engagement in physical activity for the treatment of hypertension, use of coping strategies among patients with cancer, and intentions to use mHealth apps for the management of multiple sclerosis). Given the centrality of self-management and treatment adherence-related behaviours, cognitions, and affect to disease progression, clinical outcomes, and quality of life [10], further investigation is warranted. Fourth, this review suggests that to fully understand the relevance and contribution of IU to patient outcomes, the construct must be contextualised. Consequently, the role of IU should be considered relative to other potentially influential constructs (e.g., resilience; see Mitmansgruber et al. [31]) and with consideration of change across time and the lifespan [72]. Overall, the present evidence precludes definitive conclusions regarding whether, and to what extent, IU may differ between conditions. The differing prospects, prognoses, and treatments associated with different conditions may all create uncertainty, and the extent of that uncertainty cannot be assumed to be equivalent across conditions or time. Consequently, future research may consider the interaction between IU and illness uncertainty [14] with consideration of time since diagnosis and stage of illness.

## Conclusions

The findings of this review suggest that IU is negatively associated with psychological well-being in individuals living with LTPHCs. There is limited evidence that IU may be similarly negatively associated with self-management and treatment adherence, and more research into this relationship is clearly warranted. Still, these findings appear consistent with a previous review of emotional and behavioural patient outcomes associated with IU in healthcare more broadly [35]. Similarly, the findings appear consistent with theoretical assumptions that the application of IU responses are coming at a long-term cost to the individual, possibly as a result of rigid responses aimed at controlling and/or avoiding unwanted internal experiences [20] such as those associated with worrying, condition-related fear, and catastrophising. Whilst IU may play a role in the adjustment to a LTPHC, evidence predominantly suggests that the presence of LTPHCs does not necessarily elevate IU. Notable evidential limitations emphasise the need for more rigorous and longitudinal research. Avenues for future investigation include examination of the varying influence of IU in different health conditions, population groups, across time and the lifespan, and relative to other psychological constructs. Further, this review highlights practical implications for therapy through the identification of consistent difficulties related to IU in individuals living with LTPHCs, which are discussed in relation to the nascent intervention research being done in this area. Identification of IU and its effects across various contexts may help researchers and healthcare professionals better understand and support individuals living with LTPHCs.

## Supporting information

**S1 Checklist.**
(DOCX)

**S2 Checklist.**
(DOCX)

**S1 File.**
(DOCX)

**S1 Table. Search terms and search strategy.**
(PDF)

## Author Contributions

**Conceptualization:** Benjamin Gibson, Benjamin A. Rosser, Jekaterina Schneider, Mark J. Forshaw.

**Formal analysis:** Benjamin Gibson, Benjamin A. Rosser, Jekaterina Schneider.

**Investigation:** Benjamin Gibson, Jekaterina Schneider.

**Methodology:** Benjamin Gibson, Benjamin A. Rosser, Jekaterina Schneider.

**Project administration:** Benjamin Gibson.

**Supervision:** Mark J. Forshaw.

**Writing – original draft:** Benjamin Gibson, Benjamin A. Rosser, Jekaterina Schneider.

**Writing – review & editing:** Benjamin Gibson, Benjamin A. Rosser, Jekaterina Schneider, Mark J. Forshaw.

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
