## [Decision Letter · Decision Letter 0]

8 Nov 2022

PONE-D-22-19000The role of uncertainty intolerance in self-management, treatment adherence, and psychological outcomes in individuals living with long-term physical health conditions: A systematic reviewPLOS ONE

Dear Dr. Forshaw,

Thank you for submitting your manuscript to PLOS ONE. After careful consideration, we feel that it has merit but does not fully meet PLOS ONE’s publication criteria as it currently stands. Therefore, we invite you to submit a revised version of the manuscript that addresses the points raised during the review process.

We look forward to receiving your revised manuscript.

Kind regards,

Chong Chen

Academic Editor

PLOS ONE

Reviewers' comments:

Reviewer's Responses to Questions

**Comments to the Author**

1. Is the manuscript technically sound, and do the data support the conclusions?

Reviewer #1: Partly

Reviewer #2: Partly

2. Has the statistical analysis been performed appropriately and rigorously? 

Reviewer #1: Yes

Reviewer #2: Yes

3. Have the authors made all data underlying the findings in their manuscript fully available?

Reviewer #1: Yes

Reviewer #2: Yes

4. Is the manuscript presented in an intelligible fashion and written in standard English?

Reviewer #1: Yes

Reviewer #2: Yes

5. Review Comments to the Author

Reviewer #1: This systematic literature review is well-written and comprehensive and in line with PROSPERO criteria. However, I have suggestions for minor revisions that are required prior to publication.

1) Results: what were the years of the search from inception?

2) The author hypothesized that intolerance of illness uncertainty would be related to self-management and treatment adherence, however few of the studies addressed self-management and the role of IU. This should be taken up in the discussion and more understanding of how IU would affect self-management should be made more explicit and the specific research that is needed given the potential impact of IU on self-management. It is unclear how self-management was defined in the papers and had a very limited focus i.e. adherence to medications in HIV and perhaps physical activity. Consequently, it is misleading in the conclusion as the author states that the review findings show that IU is negatively associated with self-management, yet the papers focused on a very limited view of self-management and this conclusion is not aligned with the studies reviewed and as such given the prominence of self-management in the title and as a focus of the review the author should be more explicit about this lack of evidence in the review; and its shortcomings.

Reviewer #2: Gibson and colleagues in the present manuscript entitled ‘The role of uncertainty intolerance in self-management, treatment adherence, and psychological outcomes in individuals living with long-term physical health conditions: A systematic review’ aimed to explore the relationship between intolerance of uncertainty and health-related outcomes, including physical symptoms, psychological ramifications (e.g., anxiety, depression, quality of life), self-management, and treatment adherence in individuals with long-term physical conditions (LTPHCs). The results of this systematic review showed that IU may present a psychological construct that is potentially able to change, despite being conceptualized as a trait characteristic, that influences patients’ response to LTPHCs.

The main strength of this paper is that it addresses an interesting and timely question, investigating the relationship of IU with psychological outcomes in individuals with LTPHCs. In general, I think the idea of this review is really interesting and the authors’ fascinating observations on this timely topic may be of interest to the readers of Plos One. However, some comments, as well as some crucial evidence that should be included to support the authors’ argumentation, need to be addressed to improve the quality of the article, its adequacy, and its readability prior to the publication in the present form. My overall judgment is to publish this article after the authors have carefully considered my suggestions below, in particular reshaping the parts of the Introduction and Discussion sections.

Please consider the following comments:

· I suggest changing the title. In my opinion, in the present form it seems to be too wordy and not enough clear and specific.

· Abstract: In my opinion, a lack of explanation of what the term ‘intolerance of uncertainty’ refers to and how this is related to mental and physical health makes the reader unable to grasp the key aspects of this review only by consulting this section. Please, consider on expanding this point.

· Introduction: The ‘Introduction’ section is well-written and nicely presented, with a good balance of descriptive text and information about the characteristics and relationship between intolerance of uncertainty (IU) and long-term physical health conditions. Nevertheless, I believe that more information about possible associations between IU and development/maintaining of mental health disorders, although just as a comorbidity, may provide a more accurate and scientific background to the topic: specifically, I would recommend focusing on discussing how altered levels of intolerance of uncertainty can lead to the overestimation of the possibility that a negative event will occur and to inability to cope, which results in maladaptive cognitive, emotional, and behavioural responses, therefore enhancing the possibility to reiterate maladaptive responses such as avoidance that influence the development and maintenance of mental disorders (https://doi.org/10.3389/fnbeh.2022.946263;
https://doi.org/10.3389/fpsyg.2021.737188;
https://doi.org/10.3389/fnbeh.2022.998714;
https://doi.org/10.3389/fpsyg.2021.737188https://doi.org/10.1016/j.tins.2022.04.003;
https://doi.org/10.1111/psyp.14122).

· In my opinion, the ‘Conclusions’ paragraph would benefit from some thoughtful as well as in-depth considerations by the authors, because as it stands, it is very descriptive but not enough theoretical as a discussion should be. Authors should make an effort, trying to explain the theoretical implication as well as the translational application of their study.

· In according to the previous comment, I would ask the authors to better define a proper ‘Limitations and future directions’ section before the end of the manuscript, in which authors can describe in detail and report all the technical issues that may be brought to the surface.

· Figures: Please, provide higher-quality image of the PRISMA flowchart of study selection, because, as it stands, the readers may have difficulty comprehending it.

Overall, I believe that this manuscript might carry important value providing evidence for the presence of a psychological construct that is potentially able to change in IU, despite being conceptualized as a trait characteristic, that influences patients’ response to LTPHCs.

I hope that, after these careful revisions, the manuscript can meet the Journal’s high standards for publication. I am available for a new round of revision of this paper.

I declare no conflict of interest regarding this manuscript.

Best regards,

Reviewer

6. PLOS authors have the option to publish the peer review history of their article (what does this mean?). If published, this will include your full peer review and any attached files.

Reviewer #1: No

Reviewer #2: No

---

## [Author Response · Author response to Decision Letter 0]

10 Jan 2023

PONE-D-22-19000

The role of uncertainty intolerance in self-management, treatment adherence, and psychological outcomes in individuals living with long-term physical health conditions: A systematic review

EDITOR’S COMMENT: Thank you for submitting your manuscript to PLOS ONE. After careful consideration, we feel that it has merit but does not fully meet PLOS ONE’s publication criteria as it currently stands. Therefore, we invite you to submit a revised version of the manuscript that addresses the points raised during the review process.

AUTHORS’ RESPONSE: Thank you for giving us the chance to submit a revised version of the manuscript. We have addressed each of the reviewers’ comments point by point below and marked any changes made in the manuscript using red coloured font. We believe these changes have substantially improved our manuscript. 

EDITOR’S COMMENT: Please ensure that your manuscript meets PLOS ONE's style requirements, including those for file naming. The PLOS ONE style templates can be found at: https://ddec1-0-en-ctp.trendmicro.com:443/wis/clicktime/v1/query?url=https%3a%2f%2fjournals.plos.org%2fplosone%2fs%2ffile%3fid%3dwjVg%2fPLOSOne%5fformatting%5fsample%5fmain%5fbody.pdf&umid=3d161a8e-ae92-42c4-93d1-165cfbe24cb6&auth=6b639a990a359ff1d6cc8761081d57748ce3c81e-d2a9ee959473544925ea545a3a8739a410c2436c and https://journals.plos.org/plosone/s/file?id=ba62/PLOSOne_formatting_sample_title_authors_affiliations.pdf.

AUTHORS’ RESPONSE: Thank you for highlighting these requirements. We have updated the manuscript files to align with these guidelines. 

Reviewer #1:

REVIEWER’S COMMENT: This systematic literature review is well-written and comprehensive and in line with PROSPERO criteria. However, I have suggestions for minor revisions that are required prior to publication.

AUTHORS’ RESPONSE: Thank you for your positive and thorough comments on our manuscript. We have addressed your suggested revisions point by point below and marked any changes made in the manuscript using red coloured font. 

REVIEWER’S COMMENT: 1) Results: what were the years of the search from inception?

AUTHORS’ RESPONSE: We did not limit searches by start date and included all papers published from database inception until 27 May 2022. We have now clarified this in the ‘Abstract’ and ‘Methods’ sections of the manuscript (please see page 7, line 152).

REVIEWER’S COMMENT: 2) The author hypothesised that intolerance of illness uncertainty would be related to self-management and treatment adherence, however few of the studies addressed self-management and the role of IU. This should be taken up in the discussion and more understanding of how IU would affect self-management should be made more explicit and the specific research that is needed given the potential impact of IU on self-management. It is unclear how self-management was defined in the papers and had a very limited focus i.e. adherence to medications in HIV and perhaps physical activity. Consequently, it is misleading in the conclusion as the author states that the review findings show that IU is negatively associated with self-management, yet the papers focused on a very limited view of self-management and this conclusion is not aligned with the studies reviewed and as such given the prominence of self-management in the title and as a focus of the review the author should be more explicit about this lack of evidence in the review; and its shortcomings.

AUTHORS’ RESPONSE: Thank you for highlighting this concern. We agree with the reviewer and have modified multiple sections of the manuscript in line with the above. 

First, we have changed the title of the manuscript, as follows: “The role of uncertainty intolerance in adjusting to long-term physical health conditions: A systematic review”. 

Second, we added the following sentences to the ‘Results’ and ‘Discussion’ sections of our manuscript: “Furthermore, the definition of self-management and how it was measured varied across studies” (please see page 18, lines 326–327) and “Furthermore, more research is required to explore how IU relates to specific self-management and treatment adherence behaviours across various LTPHCs” (please see page 23, lines 449–450). 

Third, we added a specific paragraph in the ‘Strengths, limitations, and future directions’ section to highlight this limitation: “Third, this review highlights that there is a serious need for more research into IU’s relationship with self-management and treatment adherence. The reviewed studies included a very limited focus of self-management and adherence (i.e., adherence to HIV medication, engagement in physical activity for the treatment of hypertension, use of coping strategies among patients with cancer, and intentions to use mHealth apps for the management of multiple sclerosis). Given the centrality of self-management and treatment adherence-related behaviours, cognitions, and affect to disease progression, clinical outcomes, and quality of life, further investigation is warranted.” (please see page 27, lines 557–565). 

Finally, we have modified the ‘Conclusion’ paragraph as follows: “There is limited evidence that IU may be similarly negatively associated with self-management and treatment adherence, and more research into this relationship is clearly warranted” (please see page 28, lines 577–579).

Reviewer #2: 

REVIEWER’S COMMENT: Gibson and colleagues in the present manuscript entitled ‘The role of uncertainty intolerance in self-management, treatment adherence, and psychological outcomes in individuals living with long-term physical health conditions: A systematic review’ aimed to explore the relationship between intolerance of uncertainty and health-related outcomes, including physical symptoms, psychological ramifications (e.g., anxiety, depression, quality of life), self-management, and treatment adherence in individuals with long-term physical conditions (LTPHCs). The results of this systematic review showed that IU may present a psychological construct that is potentially able to change, despite being conceptualised as a trait characteristic, that influences patients’ response to LTPHCs.

The main strength of this paper is that it addresses an interesting and timely question, investigating the relationship of IU with psychological outcomes in individuals with LTPHCs. In general, I think the idea of this review is really interesting and the authors’ fascinating observations on this timely topic may be of interest to the readers of Plos One. However, some comments, as well as some crucial evidence that should be included to support the authors’ argumentation, need to be addressed to improve the quality of the article, its adequacy, and its readability prior to the publication in the present form. My overall judgement is to publish this article after the authors have carefully considered my suggestions below, in particular reshaping the parts of the Introduction and Discussion sections. Please consider the following comments:

AUTHORS’ RESPONSE: Thank you for your positive and thorough comments on our manuscript. We have addressed your suggested revisions point by point below and marked any changes made in the manuscript using red coloured font. 

REVIEWER’S COMMENT: I suggest changing the title. In my opinion, in the present form it seems to be too wordy and not clear enough and specific.

AUTHORS’ RESPONSE: Thank you for highlighting this. We have changed the title to be more concise as follows: “The role of uncertainty intolerance in adjusting to long-term physical health conditions: A systematic review”. 

REVIEWER’S COMMENT: Abstract: In my opinion, a lack of explanation of what the term ‘intolerance of uncertainty’ refers to and how this is related to mental and physical health makes the reader unable to grasp the key aspects of this review only by consulting this section. Please, consider expanding on this point.

AUTHORS’ RESPONSE: We agree with the reviewer and have now added a definition of intolerance of uncertainty to the manuscript abstract: “An individual’s dispositional ability to tolerate uncertainty—or difficulty to endure the unknown—is termed intolerance of uncertainty (IU), and may play a pivotal role in their adjustment to a LTPHC.” (please see the manuscript ‘Abstract’). 

Additionally, we have expanded on how intolerance of uncertainty relates to mental and physical health outcomes in the ‘Introduction’ section of the manuscript (see the below response). 

REVIEWER’S COMMENT: Introduction: The ‘Introduction’ section is well-written and nicely presented, with a good balance of descriptive text and information about the characteristics and relationship between intolerance of uncertainty (IU) and long-term physical health conditions. Nevertheless, I believe that more information about possible associations between IU and development/maintaining of mental health disorders, although just as a comorbidity, may provide a more accurate and scientific background to the topic: specifically, I would recommend focusing on discussing how altered levels of intolerance of uncertainty can lead to the overestimation of the possibility that a negative event will occur and to inability to cope, which results in maladaptive cognitive, emotional, and behavioural responses, therefore enhancing the possibility to reiterate maladaptive responses such as avoidance that influence the development and maintenance of mental disorders.

AUTHORS’ RESPONSE: Thank you for this suggestion. We have now expanded on this in the ‘Introduction’ section as follows: “For example, IU has been found to be associated with frequent and rigid avoidance behaviours. These responses, which are aimed at controlling and/or avoiding unwanted internal experiences, appear to be a consistent feature of multiple psychological difficulties. Whilst the application of rigid avoidance behaviours may be reinforced by short-term relief, they come at a long-term cost in that they may maintain and exacerbate difficulty by restricting an individual’s behavioural repertoire at the expense of engagement in personally valued areas of life. Consequently, these strategies may paradoxically increase the unwanted experience one is seeking to avoid.” (please see page 4, lines 76–84).

REVIEWER’S COMMENT: In my opinion, the ‘Conclusions’ paragraph would benefit from some thoughtful as well as in-depth considerations by the authors, because as it stands, it is very descriptive but not enough theoretical as a discussion should be. Authors should make an effort, trying to explain the theoretical implication as well as the translational application of their study.

AUTHORS’ RESPONSE: We have made reference to the theoretical considerations in the ‘Discussion’ section of the manuscript (please see page 21, lines 401–402). Additionally, we have revised the ‘Conclusions’ paragraph to include theoretical considerations: “Similarly, the findings appear consistent with theoretical assumptions that the application of IU responses are coming at a long-term cost to the individual, possibly as a result of rigid responses aimed at controlling and/or avoiding unwanted internal experiences such as those associated with worrying, condition-related fear, and catastrophising.” (please see page 28, lines 581–585). 

We now discuss the theoretical and research implications of our work: “Notable evidential limitations emphasise the need for more rigorous and longitudinal research. Avenues for future investigation include examination of the varying influence of IU in different health conditions, population groups, across time and the lifespan, and relative to other psychological constructs.” (please see page 28, lines 586–590), as well as the practical and clinical implications: “Further, this review highlights practical implications for therapy through the identification of consistent difficulties related to IU in individuals living with LTPHCs, which are discussed in relation to the nascent intervention research being done in this area” (please see page 28, lines 590–594). 

REVIEWER’S COMMENT: In according to the previous comment, I would ask the authors to better define a proper ‘Limitations and future directions’ section before the end of the manuscript, in which authors can describe in detail and report all the technical issues that may be brought to the surface.

AUTHORS’ RESPONSE: We have included a substantial ‘Strengths, limitations, and future directions’ subsection in our manuscript within the ‘Discussion’ section (please see pages 25–28, lines 504–574). Additionally, we have now expanded on the limitations section in line with the Reviewers’ comments to include the following: “Third, this review highlights that there is a serious need for more research into IU’s relationship with self-management and treatment adherence. The reviewed studies included a very limited focus of self-management and adherence (i.e., adherence to HIV medication, engagement in physical activity for the treatment of hypertension, use of coping strategies among patients with cancer, and intentions to use mHealth apps for the management of multiple sclerosis). Given the centrality of self-management and treatment adherence-related behaviours, cognitions, and affect to disease progression, clinical outcomes, and quality of life, further investigation is warranted.” (please see page 27, lines 557–565). 

REVIEWER’S COMMENT: Figures: Please, provide higher-quality image of the PRISMA flowchart of study selection, because, as it stands, the readers may have difficulty comprehending it.

AUTHORS’ RESPONSE: Thank you for highlighting this. We have now provided a higher-quality image of the PRISMA flowchart.

REVIEWER’S COMMENT: Overall, I believe that this manuscript might carry important value providing evidence for the presence of a psychological construct that is potentially able to change in IU, despite being conceptualised as a trait characteristic, that influences patients’ response to LTPHCs. I hope that, after these careful revisions, the manuscript can meet the Journal’s high standards for publication. I am available for a new round of revision of this paper.

AUTHORS’ RESPONSE: Thank you for your positive comments on our manuscript and helpful suggestions for revisions. We hope we have addressed these satisfactorily.

---

## [Decision Letter · Decision Letter 1]

11 May 2023

The role of uncertainty intolerance in adjusting to long-term physical health conditions: A systematic review

PONE-D-22-19000R1

Dear Dr. Forshaw,

We’re pleased to inform you that your manuscript has been judged scientifically suitable for publication and will be formally accepted for publication once it meets all outstanding technical requirements.

Kind regards,

Gian Mauro Manzoni, Ph.D., Psy.D.

Academic Editor

PLOS ONE

**Comments to the Author**

1. If the authors have adequately addressed your comments raised in a previous round of review and you feel that this manuscript is now acceptable for publication, you may indicate that here to bypass the “Comments to the Author” section, enter your conflict of interest statement in the “Confidential to Editor” section, and submit your "Accept" recommendation.

Reviewer #1: All comments have been addressed

Reviewer #2: All comments have been addressed

2. Is the manuscript technically sound, and do the data support the conclusions?

Reviewer #1: Yes

Reviewer #2: Yes

3. Has the statistical analysis been performed appropriately and rigorously? 

Reviewer #1: N/A

Reviewer #2: Yes

4. Have the authors made all data underlying the findings in their manuscript fully available?

Reviewer #1: Yes

Reviewer #2: Yes

5. Is the manuscript presented in an intelligible fashion and written in standard English?

Reviewer #1: Yes

Reviewer #2: Yes

6. Review Comments to the Author

Reviewer #1: The article is clearly written and the author has made the appropriate revisions. The revision to the title is clear and the limitations section and discussion are clear and further elaborated such that the contribution of the review to the existing empirical literature is clear.

Reviewer #2: The authors did an excellent job clarifying all the questions I have raised in my previous round of review. Currently, this paper is a well-written, timely piece of research that improves the understanding of the relationship between intolerance of uncertainty and health-related outcomes, including physical symptoms, psychological ramifications (e.g., anxiety, depression, quality of life), self-management, and treatment adherence in individuals with long-term physical conditions (LTPHCs).

Overall, this is a timely and needed work. It is well-researched and nicely written. I believe that this paper does not need a further revision, therefore the manuscript meets the Journal’s high standards for publication.

I am always available for other reviews of such interesting and important articles.

Thank You for your work,

Reviewer

7. PLOS authors have the option to publish the peer review history of their article (what does this mean?). If published, this will include your full peer review and any attached files.

Reviewer #1: No

Reviewer #2: **Yes: **Simone Battaglia

---

## [Editor Report · Acceptance letter]

23 May 2023

PONE-D-22-19000R1 

The role of uncertainty intolerance in adjusting to long-term physical health conditions: A systematic review 

Dear Dr. Forshaw:

I'm pleased to inform you that your manuscript has been deemed suitable for publication in PLOS ONE. Congratulations! Your manuscript is now with our production department. 

Kind regards, 

on behalf of

Prof. Gian Mauro Manzoni 

Academic Editor

PLOS ONE